# About Model Validation in Bioprocessing

**Vignesh Rajamanickam [1], Heiko Babel [2] , Liliana Montano-Herrera [3], Alireza Ehsani [3] , Fabian Stiefel [3], Stefan Haider [1], Beate Presser [3] and Bettina Knapp [3],***

1. Boehringer Ingelheim RCV GmbH & Co KG, Biopharmaceuticals Austria, Dr. Boehringer Gasse 5-11, A-1121 Vienna, Austria; vignesh.rajamanickam@boehringer-ingelheim.com (V.R.); stefan.haider@boehringer-ingelheim.com (S.H.)
2. Boehringer Ingelheim Pharma GmbH & Co.KG, Biopharmaceuticals Germany, Birkendorfer Strasse 65, D-88397 Biberach an der Riß, Germany; heiko.babel@boehringer-ingelheim.com
3. Boehringer Ingelheim Pharma GmbH & Co.KG, Development Biologicals Germany, Birkendorfer Strasse 65, D-88397 Biberach an der Riß, Germany; liliana.montano_herrera@boehringer-ingelheim.com (L.M.-H.); alireza.ehsani@boehringer-ingelheim.com (A.E.); fabian.stiefel@boehringer-ingelheim.com (F.S.); beate.presser@boehringer-ingelheim.com (B.P.)
* Correspondence: bettina.knapp@boehringer-ingelheim.com; Tel.: +49-7351-54-188565

**Abstract:** In bioprocess engineering the Qualtiy by Design (QbD) initiative encourages the use of models to define design spaces. However, clear guidelines on how models for QbD are validated are still missing. In this review we provide a comprehensive overview of the validation methods, mathematical approaches, and metrics currently applied in bioprocess modeling. The methods cover analytics for data used for modeling, model training and selection, measures for predictiveness, and model uncertainties. We point out the general issues in model validation and calibration for different types of models and put this into the context of existing health authority recommendations. This review provides a starting point for developing a guide for model validation approaches. There is no one-fits-all approach, but this review should help to identify the best fitting validation method, or combination of methods, for the specific task and the type of bioprocess model that is being developed.

**Keywords:** bioprocess models; model validation; model calibration; quality by design; mechanistical and statistical models; hybrid models; chemometric models; biopharmaceutical engineering; regulatory guidance

## 1. Introduction

During the last few years, the biopharmaceutical industry has aimed at developing biopharmaceutical products and the corresponding process in a quality by design (QbD) manner, instead of using a quality by testing (QbT) approach [1]. Process analytical technology (PAT) can be defined as a mechanism to design, analyze, and control pharmaceutical manufacturing through the measurement of the critical process parameters (CPPs), which affect the critical quality attributes (CQAs). PAT initiatives have also been proposed by the regulatory authorities to enhance process understanding and control [2]. In QbD and PAT, the ultimate goal is to gain model predictive control (MPC) of the process to improve process performance and control of CQAs by advanced monitoring and control (AM&C) of key process parameters (KPPs) and CPPs. CPPs are, according to ICH Q8, process parameters whose variability within defined ranges has an influence on one or many CQAs [3]. KPPs show an influence on process performance parameters. To ensure that the active pharmaceutical ingredient is produced with the desired quality and performance, these parameters have to be monitored or controlled. To do this, well designed measurements (e.g., in a design of experiments (DoE) workflow) of KPPs and CPPs are necessary, as well as a corresponding process model that describes the dependencies between CQAs and the process parameters. During upstream processing online sensors or offline measurements

address the monitoring and testing preconditions. One possibility of upstream online monitoring in procaryotes is via online sensors, e.g., for turbidity or metabolite probes (e.g., with Raman [4,5] or MIR [4,6]). However, the dynamic behavior of the cells during upstream processing and the dependencies of the parameters are, especially in mammalian cell culture, rarely understood in detail. Many approaches exist for describing and modeling the biological behavior of cells in upstream biopharmaceutical manufacturing by using small experimental based models (e.g., in combinations with DoEs [7–9]), mechanistic models [10], or big data driven models, such as machine learning models [11,12], as well as partly in combinations, such as in hybrid models [13,14]. However, the quality of the models, e.g., in terms of predictability and interpretability has to be evaluated early on. This ensures that later, in the context of commercialization, the models are also well validated to make decisions about the models, such as the classification of parameters into KPP or CPP, or the definition of ranges. So far, no clear recommendations have been outlined for model validation and a straightforward and comprehensive workflow is difficult to define. One reason for this might be the diversity of mathematics (e.g., statistical, mechanistical, hybrid, etc.) and the different nature of the underlying data (scale differences, batch versus perfusion mode, sample size differences, and many more). There are no gold standard data sets available, as the biology is so diverse (e.g., different host cell lines, different targets, and so on). Thus, there is generally no clear protocol for bioprocess models available, which in turn leads to a large diversity of model validation methods [15–17].

In this review, in Section 2 we describe model validation approaches which are especially used during upstream processing in the biopharmaceutical industry. We further discuss the challenges and points to be considered when performing model validation in Section 3. These challenges might arise from the type of the underlying data, the state of the model (e.g., model training and model selection), or the risk of overfitting. Furthermore, in Section 4, we state the regulatory view on model validation methods, which is in addition to the diversity of academic research referring to a few methods only. Section 5 concludes by a final discussion and summary of the topic.

## 2. Model Validation Methods

In this review paper we focus primarily on three group of models: (1) statistical and chemometric models, (2) mechanistic models, and (3) hybrid models. For the first group, often design of experiment (DoE) data is used to make statistical models (e.g., response surface models), which describe, e.g., the relation of input parameters (i.e., factors such as CPPs and KPPs) on output parameters (i.e., responses such as CQAs), or which are used to find an optimum of a certain parameter (e.g., yield). The basis is usually a very small and limited set of experimental data. One-factor-at-time (OFATs) experiments, which have a sufficiently high statistical power (e.g., above 80%), can be used to model the relationship of CPPs/KPPs to CQAs or process performance parameters. Currently, the validation of such models is mostly performed via validation experiments [18–20]. However, this is not within the scope of the QbD approach and implies that a variety of experimental data would be required to set up the model, depending on the experimental design and the number of parameters to be evaluated.

Mechanistic models have been developed for different purposes and with different degrees of complexity, ranging from simple systems of ordinary differential equations to genome-scale metabolic network models [21]. So-called unstructured mechanistic models describe cellular processes as a black-box and balance the conversion of metabolites into cells and products in the bioreactor using systems of ordinary differential equations [22,23]. Here, the conversion and growth rates are modeled according to known or hypothesized mechanisms. These models can be naturally extended to also balance intracellular processes by the addition of intracellular compartments. In contrast to unstructured mechanistic models, which are dynamic, metabolic network models are often static and valid for a distinct time-period during the bioprocess. Here, the steady-state solutions of networks, ranging from central-carbon metabolism to "genome"-scale, are analyzed [24,25]. The

underlying assumption for these flux models is an intracellular pseudo-steady state, i.e., that intracellular conversion rates are much faster than the growth rate or extracellular exchange rates.

Hybrid (semi-parametric) models combine statistical and mechanistic models for describing a system of study [26,27]. They can be used when the process is too complex to be described mechanistically or when the process data is insufficient for data-driven approaches such as response surface models. The mechanistic part of the model has a fixed structure given by knowledge, while the other parts do not usually have a fixed structure but instead a flexible one which is determined by experimental data [13,28,29]. The advantage of a hybrid model is using data to fill or improve knowledge gaps in first principles or mechanics. Drawbacks for the implementation of hybrid models include the difficulty of establishing algorithms for the parameter identification, which are error prone and laborious. However, once a general hybrid modeling framework is implemented it is possible to reuse it for other processes and products [30].

The validation of any model should depend on the purpose of the model and the type of model used. One purpose might be to understand the dependencies of input parameters on output parameters to define KPPs and CPPs. However, sometimes the dependencies cannot be understood and modeled in detail, but still it is possible to make predictions for future batches. This is useful when, e.g., computing the probability of having out of specification (OOS) runs in future manufacturing.

As mentioned above, the simplest way to validate models is based on data (data-driven validation). This is currently the most often used method, as it is also accepted by health authorities (see Section 4). The methods which are widely used during validation of upstream models are listed and described in the following Sections 2.1–2.10. Section 2.11 gives a summary of the model validation methods.

### 2.1. $R^2$ (the Coefficient of Determination) and RMSE (Root Mean Squared Error)

The coefficient of determination ($R^2$) is in its most general definition computed by [31]:

$$R^2 = 1 - \frac{SS_{res}}{SS_{tot}}$$

with $SS_{res}$ being the sum of squares of residuals for measurements $y_i$ and mean of observed data ($\overline{y}$):

$$SS_{res} = \sum_i (y_i - \overline{y})^2$$

and $SS_{tot}$ being the total sum of squares of residuals:

$$SS_{tot} = \sum_i (y_i - f(y_i))^2$$

where $f(y_i)$ is the model at point $y_i$. $R^2$ is commonly used in statistical models, e.g., in response surface models generated with data performed during DoE runs to determine whether a model is adequate or not. To avoid overfitting, one should use the $R^2_{adjusted}$, which adjusts for the number of explanatory terms in the model in relation to the number of data points, as the $R^2$ increases with the increasing number of factors in the model. The $R^2_{predicted}$ is computed by using the model for predictions of data which have not been used in training the model.

$R^2$ is more independent than RMSE since it does not depend on the unit, and thus it can also be used to compare models trained on different data sets. Nevertheless, the $R^2$ should never be looked at independently, and the relation with the unexplained variance should be considered (see Section 3.2). There might be models which have high $R^2$ and $R^2_{adjusted}$ values, thus, they describe a lot of variance in the data, but where the RMSE is far too high (e.g., in relation to historical data or method variability). This may indicate that some effects have been missed. On the other hand, if the RMSE is too small, overfitting might

be a problem (see Section 3.2). To determine whether the RMSE is reasonable, analytical method variation, e.g., derived from control charts or historical data, can be used.

The RMSE is computed by taking the square root of the mean square error [32]:

$$\text{RMSE} = \sqrt{\frac{\sum_{i=1}^{n}(f(y_i) - y_i)^2}{dof}}$$

with *dof* being the degrees of freedom. The RMSE is usually used to judge the performance of the trained model and to analyze the predictive power of the model, e.g., with a validation dataset.

The RMSE of validation should be compared to the measurement error (reference method, reproducibility error) to avoid overfitting/underfitting (see Section 3.2).

Alternatively to the RMSE, the mean absolute error of prediction (MAE) can be used [32]. If divided by the standard deviation of the experimental values the normalized MAE (nMAE) and normalized RMSE (nRMSE) are unbiased measurements for model predictions.

### 2.2. Accuracy (Closeness of Prediction to Real Value) and Precision (Random Error of Model Predictions Comparable with Reproducibility of Real Data)

For models which categorize results into positive and negative the accuracy is computed as (TP + TN)/(TP + TN + FP + FN) [33] and precision (sometimes also called positive predictive value) as TP/(TP + FP) [34]. For continuous data which cannot be categorized into groups, accuracy should ideally be assessed by comparing the results obtained with the computational method (simulated in vitro data) with the results of an independent set of real data (not used in training for example historical data). Then, for both data sets accuracy and precision can be computed (e.g., by computing the $R^2$ and the RMSE) and an acceptance criterion (computed, e.g., on historical data) can be determined to see whether the results are comparable. For analytical procedures there are different layers of precision (reproducibility, intermediate precision, analysis repeatability (replicates), systems repeatability (repeats) [35]). For computational models there are no such layers; however, one should be aware of the precision which will be reached by the model. Repeats have a narrower standard deviation than the reproducibility (e.g., long-term measurements). Moreover, not every model will aim at the same precision depending on its application; according to Box: "all models are wrong, but some are useful. However, the approximate nature of the model must always be borne in mind...." [36].

### 2.3. Specificity and Sensitivity (True Negatives and True Positives) and ROC

Classification models, not used for predicting process variables, give a qualitative overview of the process performance and can be used to, e.g., identify different phases in a process. Spectroscopic sensor data are used to build such classification into chemometric models to monitor the process performance. The performance of such models is assessed with true negative and true positive rates (TNR and TPR) using internal crossvalidation (see Section 2.4) and external validation approaches [37]. True positive and true negative means that the model classifies the observation into the class where it actually belongs. The TPR is computed as TP/(TP + FN), where TP are true positives and FN are false negatives. TNR is computed as TN/(TN + FP), where TN are true negatives (positive events wrongly categorized as negative) and FP are false positives (negative events wrongly categorized as positive). The false positive rate (FPR = FP/(FP + TN)) is the ratio of the number of FP and the total number of actual negative events (regardless of classification). Plotting the TPR against the FPR results in a receiver operating characteristic (ROC) curve and illustrates the ability of the classification if the discrimination is varied. The area under the ROC curve (AUROC) is perfect if it is equal to one and only as good as a random classification if it is equal to 0.5.

Specificity and sensitivity are measures for the proportion of true negatives and true positives, respectively, that are correctly identified by the model. Thus, it is similar to the

TNR and TPR measures. The sensitivity and specificity can be used for binary outcomes or classifications models, for example, in equivalence testing [38].

*2.4. Crossvalidation, Such as Leave-One-Out Crossvalidation (LOOCV) and Leave-Multiple-Out CV*

Crossvalidation (CV) is an internal re-sampling method, where the original dataset is split into training and validation datasets. These datasets are simultaneously used for model building and validation. The model is trained or built with the training dataset by leaving out a part of the original dataset. The validation dataset can be a part of the original dataset or an external dataset used only for validating the model. The trained model is then used to predict the responses based on the validation dataset [39]. CV is often applied to smaller datasets (e.g., sample size < 10) during initial model development and validation. CV at different levels provides important insights regarding model validation (e.g., source of variation, comparability) and for assessing the main sources of variation. It is important with CV to include samples at different levels (e.g., scales) in the training and validation datasets to ensure model reliability. A validated $R^2$ (see Section 2.1) from a smaller CV sample (e.g., LOOCV) is necessary for the initial training validation but not sufficient to ensure predictive performance in the validation and implementation stages [40]. Especially with chemometric and statistical models, the underlying stratification, from splitting the original dataset into training and validation, plays an important role and must be taken into account for remediation and improving the model performance.

Monte Carlo CV (MCCV) has been shown in applications to be more consistent in comparison to conventional CV approaches [41]. The authors compared three different methods, namely MCCV, LOOCV, and k-fold CV, to determine the optimal number of model variables achieving a predetermined prediction accuracy. Model validation using LOOCV renders an unnecessarily high number of model variables, due to overfitting, which ultimately reduced the predictive capability of the model. k-fold CV is a procedure for determining the accuracy of the model on new data. The procedure has a single parameter 'k' that refers to the number of groups into which a given dataset is split. Furthermore, when k-fold CV is used, the computational power increases exponentially to determine the optimal number of model variables to achieve the predetermined prediction accuracy. Optimization of model parameters has been recently proposed using a two-layered (internal and external) cross validation approach for chemometric models [42]. The model parameters are optimized using an internal CV approach, whereas the generalized model evaluation was done using an external CV approach. Since preprocessing is a crucial step in chemometric model development, preprocessing parameters can also be optimized inside the internal CV iterative loop. Other re-sampling approaches, such as Jacknife, holdout, and bootstrapping, were also tested and compared to the two-layered CV approach. Finally, k-fold and k-replicate CVs were used to analyze the difference between the calibration and test sets and to account for the reproducibility between replicate samples [42].

For selecting subsets for model calibration and validation for leave-multiple-out cross-validation there are several resampling methods, namely the Kennard and Stone algorithm (maximin criterion) [43], Duplex (modification of Kennard Stone) [44], D-optimality criterion (maximise determinant of the information matrix) [45], and K-means or Kohonen mapping (the latter is extensively used in neural networks) [46]. The resampling methods determine how the original dataset is split for validation. A comparison of various resampling methods has been reviewed elsewhere [47,48].

*2.5. Repeatability, Intermediate Precision*

In validation of analytical procedures quality testing is necessary to confirm that the analytical procedure is suitable for its intended use. Similarly, the validation of a computational model should confirm that the model is valid for describing the intended problem at hand [49,50].

For the analytical validation, the intermediate precision and repeatability (intra-assays precision) have to be determined. The intermediate precision of analytical procedures considers several different factors, such as date of test, test analyst, apparatus, etc. In terms of a computational model this relates to factors such as the start parameters, data (data split in CV), and so on. Repeatability should be done independently, for example using CV. Therefore, one can have homogenous samples and also heterogeneous samples. Replicates within the underlying data should especially be handled with care (e.g., use all replicates, either within the test or within the validation set in CV).

### 2.6. Maximum Likelihood

Likelihood is the probability density function that a dataset is observed given a parameter-set $\theta$ [12]:

$$\mathcal{L}(\theta \,|x) = p_\theta(x) = P_\theta(X = x)$$

wherein $X$ is a random variable with probability mass function $p$ depending on $\theta$.

The formulation of a likelihood limits its application to parametric models, however there are extensions dealing with non-parametric likelihood approaches [51]. The likelihood function is the basis for objective functions and used to derive the so-called maximum likelihood estimator [52]. Under the assumption of independent and normal distributed error terms this directly leads to the residual sum of squares (RSS), weighted by the variance of the error term [53]. The maximum likelihood estimator is a widely used objective function for bioprocess models. The obtained likelihood value for a model with a dataset can be used for model comparison. Here, the likelihood ratio test is performed (see likelihood ratio), but the Akaike information criterion (AIC) (see Section 2.6) can also be used. Likelihood values are also computed for model diagnostics [22] and for model uncertainty (see Section 2.9).

### 2.7. Information Criteria (Akaike Information Criterion (AIC), Bayesian Information Criterion (BIC))

When fitting a mechanistic or statistical model, it is possible to increase the model fit by adding parameters, which can result in overfitting. In this sense, AIC, AICc (corrected AIC used for small sample sizes) [54], and BIC are the most widely used selection criteria for the modeling and identification of upstream systems to achieve the simplest model with the least variables but with greatest explanatory power. Both AIC [55] and BIC measure the trade-off of model fit [56] (quantified in terms of the log-likelihood (see Section 2.6)) with model complexity (a penalty for using the sample data to estimate the model parameters):

$$AIC = -2logL + 2K BIC = -2\log L + K \log N$$

where $L$ is the likelihood, $K$ is the number of model parameters, and $N$ is the number of data points used to train a model (computed on the joint training/validation data). A model with better fit has smaller AIC or BIC, and while AIC and BIC penalize a model for having many parameters, BIC penalizes a model more severely compared to AIC (for $N > \exp(2)$) with the increasing quantity of data [57–59]. Therefore, BIC could be more suitable for selecting a correct model, while the AIC is more apt for finding the best model for predicting future observations for a given data set.

### 2.8. Goodness-of-Fit

The goodness-of-fit (GOF) of a set of results of a model describes how this set of simulated results fits the observation dataset. When multiple models of a process are available, the GOF gives an assessment of relative model fit and provides information on selecting the superior model. Therefore, in the context of model validation the GOF can be used for two purposes: (i) validation of the simulation results of a single model, and (ii) relative validation of different models' simulations.

The two most popular standard GOF statistics are Pearson's statistics ($\chi^2$) [60],

$$\chi^2 = \sum_{c=1}^{C} \frac{(p_c - \hat{\pi}_c)^2}{\hat{\pi}_c}$$

and the likelihood ratio,

$$G^2 = 2N \sum_{c=1}^{C} p_c ln \frac{(p_c)}{\hat{\pi}_c}$$

where $c$ is the contingency table, $\pi_c$ is the probability of the $c$, $p_c$ is the observed proportion, and $\hat{\pi}_c$ is the probability of the cell $c$ under the model.

In order to use the GOF methods in model selection studies, the two most popular GOF indices, AIC and BIC can be used (Section 2.7).

### 2.9. Model Uncertainty, Model Robustness

There exists a parallel between the uncertainty about the data and the uncertainty about the model and its predictions (see Section 3.4). The typical standard errors and confidence intervals indicate uncertainty about the data, measuring how an estimate changes with sampling. However, a robust model should also account for model structure and its predictive capability in terms of nonlinear effects and heterogeneities.

Data collection in upstream processes can be noisy and susceptible to errors (i.e., corrupted sensors, errors in the measurement devices, etc.). Standard statistical models and data analytic techniques can fail under such scenarios, which can reduce their applicability. A robust model should handle various forms of errors, as well as changes in the underlying data distribution in an automatic way, allowing models that can be used in a reliable way, and enabling their employment even in complex applications such as biopharmaceutical bioprocess.

Many of the methods for characterizing uncertainty in models apply equally to all types of models. However, different assumptions are made and might be applicable for a certain model type and a certain dataset. In general, different approaches are utilized:

- Linear approximations of confidence intervals: These methods rely on the numerical estimation of a Jacobi matrix of the model with respect to the parameters (or weights for artificial neural networks (ANNs)). Uncertainties in the training data can directly be propagated by linear approximation [17,61].
- Bayesian approaches: Here, the uncertainty in estimated parameters needs to be determined first (e.g., using likelihood approaches). Model ensembles or distributions of outputs are then obtained by sampling the multi-dimensional parameter distributions using, for example, Markov-chain Monte Carlo methods [15,62].
- Bootstrapping: Here, the original data for model-training is bootstrapped. This results in a model ensemble that produces an output distribution that depends directly on the data uncertainty [63].
- Mean Variance Estimation (MVE) Method: This method is unique to ANNs. Here, the ANN is trained to learn an additional output, which is the uncertainty in the prediction [17,61].
- Validation Profile Likelihood: This method is based on the maximum likelihood estimator (see Section 2.6). Here, likelihood values of hypothetical data-points are calculated, and using the $\chi^2$ distribution, a confidence interval with level $\alpha$ can be determined [64].

### 2.10. Credibility Score and Continuous Testing

Model credibility accounts for the risk associated with the decisions made using the computational model. Based on a risk assessment, the quantitative and qualitative levels of credibility that need to be achieved have to be determined prior to model building. This is a so called risk-informed credibility assessment framework and is used in the American Society of Mechanical Engineers (ASME) norm for computational models of medical

devices [65]. It operates in line with in vitro (e.g., bench testing) and/or in vivo testing (e.g., experiments) to demonstrate the validity of the predictions of the computational model. Model credibility aims at the establishment of trust in the predictive capability of the model. To collect evidence from the credibility activities and to establish this trust, verification studies of the code and the calculations are one part. Another part is the comparability assessment with the test samples (in vitro or in vivo) by looking at equivalences of the inputs and comparison of the outputs. Although developed for physics-based models, the model credibility concept can be extended to other types of models, such as statistical and machine learning models, as well as their application in pharmaceutical and biological products [66,67]. However, the approach as defined in the ASME norm is very regulated and may cause problems for models of biological systems which show high variability, such as that seen in bioprocesses [68].

### 2.11. Summary of Validation Method

In Figure 1 the model validation methods are summarized and an overview is given. Basically, four points should be considered when deciding which methods should be used for model validation:

- Nature of the dataset: Are the data representative? Are there replicates given in the data? Is the variation in the data high (e.g., does the DoE allow modeling the full design space or are only runs with the same settings available)?
- Sample size: Is the dataset used of low or high sample size (e.g., more than 10 samples)?
- Model state: In which state is the model, i.e., model selection (e.g., identify whether a linear model versus a quadratic model should be used), training (e.g., perform the linear fit), or implementation (e.g., make a linear fit during each campaign of commercial manufacturing)?
- Model type: Is the model a statistical, a mechanistical, or a hybrid model?

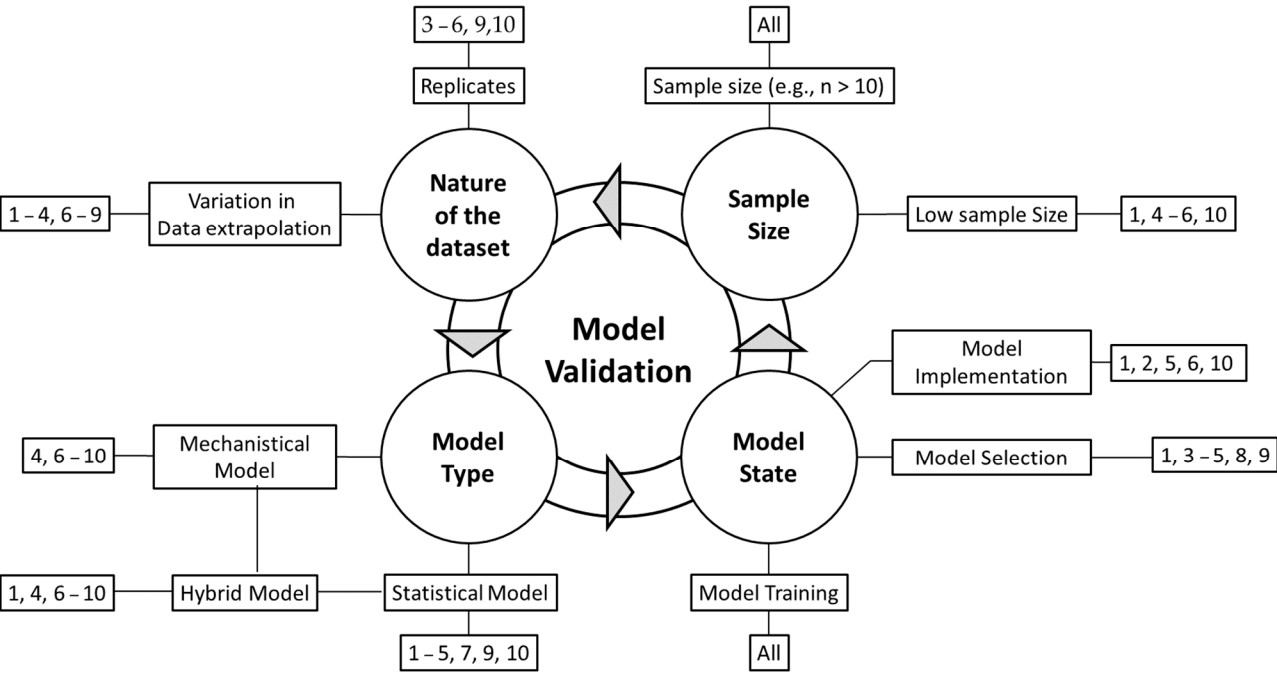

**Figure 1.** An overview of the different aspects to consider when choosing a validation statistic or method. Section 2 sub-headers, namely Sections 2.1–2.10 are denoted as 1–10.

All of these four points are interconnected. The definition of what a "low" and "high" sample size number means depends on the other three points. For example, for the selection of a mechanistic model, higher samples sizes (e.g., $n > 10$) might be more necessary than

for the selection of statistical models. However, for model training, mechanistic models may need lower sample sizes than statistical models. On the other hand, if there is a high sample size given, but if all measurements are performed with the same settings, thus only replicates of a single experiment are given, this will not be helpful for model training. Thus, there is not a one-fits-all approach, and the definition of a single workflow is probably hard to give. Furthermore, the comparison of different models and their validation is difficult, due to, for example, the lack of gold standard data sets. Nevertheless, this summary should show that there is a plethora of different validation methods, and that for each model it should be critically assessed which method or combination of methods is best suited for validation. We summarized some important aspects in the context of bioprocessing which should be taken into account and which might help in the choice of choosing validation methods in the next Section.

## 3. Further Points to Consider

In this Section, we give advice about what should be taken into account during model validation for bioprocesses. As described in Section 2.11, a clear validation protocol is hard to define and depends on the underlying data, type, and state of the model. Therefore, the following aspects can be helpful for deciding which methods should be used.

### 3.1. Calibration/Model Fitting

Model calibration plays a vital role in model validation and the future of use of the calibrated model for prediction. The validation statistics can be interpreted only when the calibrated model and the data used for calibration are robust. Well calibrated models can be useful tools for studying the underlying mechanisms and rationalizing the results [12]. However, the techniques used in model calibration also vary widely, such as, subset or feature selection and data splitting approaches, depending on the type of model (e.g., mechanistic vs. chemometric). Therefore, clear calibration and validation protocols are not available [69]. The data needed to calibrate and validate a model is heavily reliant on the type of model in question. For example, a mechanistic model would require a much smaller dataset in comparison to a hybrid [13] or chemometric model [70]. Therefore, validation statistics and approaches are also dependent on the sample size of the datasets during model development. Nevertheless, the use of sound validation statistics at each level of the model calibration workflow would pave way for overall applicability and streamline the model calibration protocol.

### 3.2. Overfitting and Underfitting:

During model validation it is important to account for underfitting (i.e., oversimplifying, e.g., by using a linear model instead of a quadratic model), but overfitting should also be avoided [46,71]. In both scenarios the model will make errors in prediction. In the case of overfitting, measurement noise could be interpreted as being a process relevant effect, e.g., by fitting a quadratic function into a truly linear process. In the case of underfitting, some truly existing process effects might be missed. For example, if a linear fit is performed but the process is actually quadratic.

In data driven approaches (e.g., machine learning) this phenomenon is captured by the so-called bias–variance tradeoff, which is a decomposition of the mean squared error into the "bias" term, which represents how well an average value is predicted, and the "variance" term, which increases when the model is overfitted [72]. To obtain statistical models with good generalization properties it is either common to apply early stopping in the learning phase [73] or to use regularization terms in the loss functions [74]. If the model is trained using cross validation, a comparison of the $R^2$ values (see Section 2.4) for the training and test data (called Q2) can be informative for determining if overfitting occurs [75–77].

### 3.3. Limit of Detections and Limit of Quantification (LoD and LoQ)

In some analytical procedures there are detection and/or quantification limits. The LoD is the lowest possible distinction from noise of the analytical method [78]. The LoQ is the lowest quantitatively measurable amount with suitable accuracy and precision for the analytical procedure [49,79]. These limits should be considered when using data to build a model, as the model might not reflect the limits, and thus bias results upon reaching a certain limit. The LoD and LoQ are computed as follows [80]:

$$\text{LoD} = 3.3 * \frac{s_{noise}}{b} \quad \text{LoQ} = 10 * \frac{s_{noise}}{b}$$

with $s_{noise}$ being the standard deviation of the calibration curve and $b$ being the slope of the regression line. In bioprocesses, often the relations of calibration curves are linear, and thus the above mentioned formulas can be used. However, this might not always be the case, depending on the type of data and underlying measurements. If a linear regression cannot be used directly, data transformation can be an option (see Section 3.4), before using the data in calibration, and also later in validation. Another option is using more complicated formulas [81,82].

### 3.4. Homogeneity of Variance

Homogeneity of variance (HOV) is an assumption of the validity of many parametric tests, such as the t-test and ANOVA, that rely on the assumption that the true population variance for each group is the same as in the observed sample. During model calibration HOV is important to compare data sets that have intrinsic differences, to decide if they are comparable and can be used together to develop a model. Testing for HOV is also useful during validation to compare two data sets that do not come from the same source, which is mostly the case for the calibration and validation data sets. For instance, in bioprocesses comparing the daily average substrate consumption rates and product formation efficiencies of two different sets of bioreactors, which were initiated using a different inoculum. If the assumption of HOV is not met, it might be problematic to use the datasets to validate a model right away. In this case, data transformation (such as log transformation) of the response variable can be helpful. The most common test to check for HOV are Levene's test [83], Bartlett's test, Brown and Forsythe's [84], the Welch test, and F-max test [85].

### 3.5. Type of Data

As part of model validation, the predictive accuracy of a model must be evaluated by comparing the model prediction against measured process data to ensure the model was built correctly. Certain modeling approaches such as statistical and chemometric models can require a large amount of data for their calibration and testing. Therefore, quality training and test data sets need to have certain features, such as enough information (variability in process inputs), a sufficient number of experiments and observations, and in the case of on-line data, a low signal to noise ratio is critical to obtain reliable models. Further aspects to consider are the sampling timing points and the process intrinsic variability. On the other hand, if a large number of measurements is given, but they show only small variations in settings (e.g., only replicates are given) then a model is hard to build [86].

Furthermore, discrete data should be treated differently from continuous data. In general, it is recommended to always use the unrounded raw values when developing and validating a model.

Biopharmaceutical upstream manufacturing largely depends on batch processes, with data sets containing time dependent information with a typical three-dimensional shape of batches x variables x time. Unfolding procedures are used to reduce this three-dimension matrix to a two-dimensional format, which is necessary for multivariate data analysis. The data set can be unfolded in different ways, depending on the purpose of the analysis. Batch-wise unfolding, where each row in the matrix is a different batch, is used to analyze differences among batches by removing the dynamic behavior of the batch. Conversely,

variable-wise unfolding is used to study the dynamic behavior of the batch relative to the mean of each variable [87,88].

Moreover, in biopharmaceutical upstream processes there are different kinds of parameters, such as input parameters, which are highly controlled in a quite narrow range and others which are controlled only indirectly within a broad range. Some parameter settings cannot be tested in real experiments as they would lead to an edge-of-failure. For some parameters, online measures give a high time resolution and thus a large set of data, for other parameters only offline measures at certain time steps are available. This has to be taken into account during data cleaning and polishing and should be an important part of the whole model lifecycle to ensure that the data fits the problem and is suited for the model.

Intensified DoEs (iDoEs) have been recently used in upstream processing to develop hybrid models in an efficient manner, with a small amount of data needed [14,29,89]. Independent of the number of samples measured, the nature of the data used for model generation also plays a role in model validation, as described above. Furthermore, if there are replicates within a dataset they should be studied in detail. In CV, for example, it is necessary to have variance in the data for the RMSE to be meaningful. The RMSE can, for instance, be very small in the training set and very large in the test set if the data sets are chosen incorrectly.

### 3.6. State of Model

Typical model states in bioprocesses are model development for generating base process knowledge, using a developed model for process monitoring, prediction and optimization activities, and, ultimately, for continuous process improvement [90]. The amount of data needed during model development stages is much higher than during the implementation and maintenance stages. Maintenance of the model necessitates that any improvements and changes to the model be continuously assessed throughout the model lifecycle; namely, through development, implementation, and further maintenance. Based on the available data that are relevant for the model state, validation statistics can vary vastly.

### 3.7. Good Modeling Practice (GMoP)

Ideally, during model validation several methods are combined to account for over- and underfitting, but also to assess different aspects of the model. For example, there might be models which have a very small RMSE but where the model predictability is rather poor due to effects which were not considered (e.g., equipment variability). Therefore, it is essential to adhere to a good modeling practice. Consequently, this will address all the aforementioned points to be considered, in context. Typically, this starts with a clear definition of an objective and the necessary requirements for a specific model type. Based on the model nature, different assumptions are discussed and model calibration is done. Thereon, the sensitivity analysis and an estimation of parameter uncertainty is performed. Adhering to GMoP, this will ensure a robust model, capable of simulating and predicting outcomes [91].

## 4. Recommendations from Health Authorities

This section describes the view of health authorities in the field of bioprocesses.

According to the ICH guidelines Q8, Q9, and Q10, model validation is an essential part of model development and implementation, and verification of such models must be carried out throughout the lifecycle of the product [92]. Furthermore, models should be categorized based on high, medium, or low impact models, and the validation extent for such models must be considered based on their level of impact on the process. The following points must be considered for high impact models, namely, setting acceptance criteria, the comparison of accuracy using internal cross validation, validation of the model using external cross validation, and verifying prediction accuracy by parallel testing with

the reference method throughout the lifecycle. Validation procedures should take into account any changes in material attributes or analytical procedures and differences arising from scales.

In a PAT framework, validation procedures should consider analytical method validation and continuous quality assurance [35]. Statistical methods such as ANOVA for assessing regression analysis (e.g., in a chemometric model), $R$ (correlation coefficient), and $R^2$ or linear regression for linearity can be used to assess validation characteristics [79]. Computational and simulation models should be described to assess their validity and their prediction capability for the model outputs with validated analytical methods [93]. The sensitivity of the model outputs on the key model parameters must be described with a systematic analysis of the uncertainty. Although different validation procedures are presented in academic research, only a few are mentioned in the regulatory documents. Intensive use of modeling approaches to accelerate product approval stages would enable the addition of robust validation approaches in regulatory documents.

Model implementation in production processes necessitates the validation of the software when the model is a part of the production process or the quality system [94]. This is especially relevant for automated data processing or analysis during the production process, where any software changes must be validated before approval and issuance. In accordance to the general principles of software validation, appropriate software design to accommodate changes (modular setup) is necessary to reduce future validation efforts.

## 5. Concluding Remarks

In this review, the validation methods currently used in bioprocess modeling are described. QbD and the necessity of speed-to-clinic approaches make modeling more and more important in the biopharmaceutical industry. Together with the fact that extensive experimental set-ups are expensive and time consuming, modeling approaches often allow a better process understanding, optimization, and control. However, the reliability of the given model needs to be ensured before answering the questions at hand (e.g., optimization of titer or the definition of KPPs and CPPs). As there are many different model types (statistical, mechanistical, hybrid, and so on) a unique method of model validation is difficult, or even impossible [95]. This is also due to the fact that there is no clear protocol for model generation in bioprocessing. Even health authorities refer to some basic approaches such as $R$ and $R^2$.

There exists a plethora of methods for model validation and in consequence of the points mentioned above, using a combination of several methods is recommended. This will better allow judging the reliability and predictability of the model. Furthermore, it helps to account for under/overfitting and to address the type of model, even throughout the modeling lifecycle, such as during model calibration, validation, and implementation. In any case, data are needed to set-up a model, and thus, the data should be checked very carefully before model generation and validation. For example, the sample size, the variation within the data, but also the number of replicates should be considered.

Ideally, beside using model validation for in silico methods, models should undergo a lifecycle which includes continuously testing during a product lifecycle, which is also recommended by health authorities [90]. Here, the purpose of the model can also be considered, e.g., using a model for having a better process understanding in early stage development, for defining CPPs in late stage development, or for process control during manufacturing. If the model has a high impact on the process, the model validation should be more detailed then for models with a lower impact. We expect that validation methods in bioprocess development will become more important as the models themselves become more widely accepted. Using all the available and suitable modeling validation approaches will help to judge the models regarding their predictability and reliability. Reliable models will help to understand bioprocesses in their entire lifecycle (lab to market) and thereby make them more robust and flexible.

**Author Contributions:** Conceptualization and writing (original draft preparation): V.R. and B.K. with important contributions from H.B., L.M.-H., A.E. and revisions at multiple stages; writing review and editing: V.R., B.K., H.B., L.M.-H., A.E., F.S., S.H. and B.P.; visualization: V.R. All authors have read and agreed to the published version of the manuscript.

**Funding:** This research received no external funding.

**Institutional Review Board Statement:** Not applicable.

**Informed Consent Statement:** Not applicable.

**Acknowledgments:** We would like to thank Christina Yassouridis, Martina Berger, Albert Paul, Ogsen Gabrielyan, Joachim Baer, Erich Bluhmki and Jens Smiatek for valuable discussions and useful hints. Furthermore, we thank the anonymous reviewers for very useful comments.

**Conflicts of Interest:** The authors declare no conflict of interest.

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
