# Peer review of "About Model Validation in Bioprocessing"

_processes, doi:10.3390/pr9060961_

Round 1

Reviewer 1 Report

The manuscript reviews the validation methods using three sections after the introduction, one for validation methods, the other for further points to consider, and the last one for recommendations from health authorities. The work is very relevant, as literature usually focuses on other modeling steps overlooking the validation step. Nevertheless, the manuscript requires major changes in the way the methods are described and structured to transmit coherent and useful information without being a mere enumeration of methods. I consider that the changes are not easy to implement as there are many different methods applied to different types of models and data, but at least the next major changes should be addressed (I include only one example for each item to illustrate the problem, but authors should carefully check the whole document for other specific cases)

  • Section 2 is an enumeration of methods (11 subsections). But there is not a clear reason for ordering the methods in the way it is done in the document. In fact, there are cases where a method is compared with others that were not yet defined. For example in section 2.4 R2 is defined and compared with RMSE and later on RMSE is defined. Could not be useful the grouping of methods based on similitudes in the type of models and data to be validated? And start describing the most basic concepts as RMSE first?
  • Some methods are mentioned and discussed without being described at all. For example line 152 " Finally, k-fold and k-replicate CVs were used to analyze the difference between the calibration and test sets and to account for reproducibility between replicate samples". Even k-fold appears in other parts of the document, but without description.
  • Some methods are described for only one type of model when it can be applied for other bioprocessing models. For example, in 2.3 accuracy and precision are described for models categorizing the outcome in positive and negative results, but it is not described for quantitative continuous data where data are not categorized in groups (in terms of variance, error...)
  • Some statements are not strictly true, for example in line 238 "Under the assumption of normal distributed error terms this directly leads to the residual sum of squares (RSS) weighted by the standard deviation of the error term". To be strict, error should be also independent and the weight is by the variance, not the standard deviation. 
  • Another example in line 251 " A model with better fit has smaller AIC or BIC, and while AIC and BIC penalize a model for having many parameters, BIC penalizes a model more severely compared to AIC" by definition this is only true if klogN>2k, i.e. for N>exp(2).
  • Not clear sometimes how the method applies to validation, example section 2.9, the same could be said when calibrating?
  • Authors used many similar terms that seem to be the same term (or not, not sure as are not defined or a clear context), what difficulties the reading. For example, in line 93 "In contrast to dynamic mechanistic models", but previous lines described unstructured mechanistic models. Therefore to make the connection with previous sentences, are all unstructured models dynamic (not completely clear for the context as ordinary differential equations can be depending on another variable that is not time)?
  • Some sentences are incomplete or seem without context (line 107 "However, once a general hybrid modelling framework is implemented it is possible to reuse it" to reuse it for what?", last paragraph in section 4 only states "Model implementation in production processes necessitates validation of the software, when the model is a part of the production process or the quality system [91]" when software was not discussed in another part of the document)
  • It should be clear the type of models and data that each method is applicable, and this is not usually described. For example in section 3.3 b is the regression line, is this method therefore only applicable to linear regressions? 
  • It is not clear the differences between sections 2 and 3. Is this section 2 referring only to validation and 3 to other aspects not affecting validation that are not validation methods? Note that, for example, the first paragraph in section 3 uses the same (or very similar) example to illustrate dependence on the type of model than in the last paragraph in section 2. But not connection is explained. 

Other minor comments.

  • Line 449 "In CV it should be avoided to have replicates within the test and training set to not artificially improve validation measures such as RMSE" Is the training set included in the validation measure with RMSE?
  • Some common acronyms are not defined (like PAT), and others are defined but difficult to know the difference from the reading of the work, such as KPPs and CPPs. Also, the intro uses too many acronyms at once, making it difficult the reading. 

Reviewer 2 Report

The paper discusses the problem of validation methods applied in bioprocess modelling. The well known fundamental description of models and the way of their calibration is served. The manuscript only repeats the generally known knowledge of statistical analysis that is applied to bioprocess modelling. Thus, I am not able to recognize the novelty of the manuscript.  

Author Response

Thank you for your feedback. With the manuscript we show the variety of available validation methods and the need of looking at validation very carefully and therefore, we think a review like this is very much needed. As far as we are aware of, a comprehensive review of different validation methods are not yet published. We also emphasize the aspects that affect model validation and regulatory agencies current demands. We consider this to be a novelty of this review manuscript.

Reviewer 3 Report

This is a well-written review article, looking at various model validation methods in bioprocessing. I would recommend the paper be accepted for publication.

Author Response

We thank the reviewer very much for the positive feedback.

Round 2

Reviewer 1 Report

I cannot find any letter to the reviewers discussing the changes in the manuscript. I only see the document with the track of changes generated directly by latex. In the current form, I cannot assess properly if the new version includes the proposed changes and where (or the justification for the changes that have not been addressed like the change of structure)

Reviewer 2 Report

I am still confused about the novelty because you present already available mathematical tools that are just applied to biotechnology. But okay, such a comprehensive review can help to unfamiliar persons to understand the problem of model validation in biotechnologies.

Author Response

Response: Thanks for your comment and we again emphasize that we address with this manuscript  current demands of regulatory agencies and the whole bioprocess field for an overview about model validation methods.

Round 3

Reviewer 1 Report

In the current form, the manuscript organization and readiness has improved considerably. 

The authors have conducted strictly all the proposed specific changes. However, in my letter, I stressed that I was proposing only one specific change as an example for each general comment, but there were other required changes. It is not the task of the reviewer to enumerate each of the very specific problems. It is unfortunate that the authors have not conducted a deeper revision, as the final manuscript could have been really interesting work.

Nevertheless, the work is appropriate for the journal and can be approved in the current form.